# Tissue-derived exosome proteomics identifies promising diagnostic biomarkers for esophageal cancer

Dingyu Rao[1†], Hua Lu[2†], Xiongwei Wang[3†], Zhonghong Lai[4], Jiali Zhang[5], Zhixian Tang[1]*

[1]Department of Cardiothoracic Surgery, First Affiliated Hospital of Gannan Medical University, Ganzhou, China; [2]The First Clinical School of Medicine of Southern Medical University, Guangzhou, China; [3]Department of Anesthesiology, Longhua District Central Hospital, Shenzhen, China; [4]Department of Traumatology, First Affiliated Hospital of Gannan Medical University, Ganzhou, China; [5]The First School of Clinical Medicine, Gannan Medical University, Ganzhou, China

*For correspondence:
tangzhixian@gmu.cn

[†]These authors contributed equally to this work

Competing interest: The authors declare that no competing interests exist.

**Abstract** Esophageal cancer (EC) is a fatal digestive disease with a poor prognosis and frequent lymphatic metastases. Nevertheless, reliable biomarkers for EC diagnosis are currently unavailable. Accordingly, we have performed a comparative proteomics analysis on cancer and paracancer tissue-derived exosomes from eight pairs of EC patients using label-free quantification proteomics profiling and have analyzed the differentially expressed proteins through bioinformatics. Furthermore, nano-flow cytometry (NanoFCM) was used to validate the candidate proteins from plasma-derived exosomes in 122 EC patients. Of the 803 differentially expressed proteins discovered in cancer and paracancer tissue-derived exosomes, 686 were up-regulated and 117 were down-regulated. Intercellular adhesion molecule-1 (CD54) was identified as an up-regulated candidate for further investigation, and its high expression in cancer tissues of EC patients was validated using immunohistochemistry, real-time quantitative PCR (RT-qPCR), and western blot analyses. In addition, plasma-derived exosome NanoFCM data from 122 EC patients concurred with our proteomic analysis. The receiver operating characteristic (ROC) analysis demonstrated that the AUC, sensitivity, and specificity values for CD54 were 0.702, 66.13%, and 71.31%, respectively, for EC diagnosis. Small interference (si)RNA was employed to silence the CD54 gene in EC cells. A series of assays, including cell counting kit-8, adhesion, wound healing, and Matrigel invasion, were performed to investigate EC viability, adhesive, migratory, and invasive abilities, respectively. The results showed that CD54 promoted EC proliferation, migration, and invasion. Collectively, tissue-derived exosomal proteomics strongly demonstrates that CD54 is a promising biomarker for EC diagnosis and a key molecule for EC development.

## Editor's evaluation

This study advances our understanding of the predictive role of tissue-derived biomarkers for esophageal cancer. The evidence supporting the conclusions is solid. However, there are a few areas in which the article may be improved through further analysis and validation of the clinical usefulness of CD54 as diagnostic biomarkers for esophageal cancer. The work will be of broad interest to clinicians, medical researchers and scientists working in esophageal cancer.

## Introduction

Esophageal cancer (EC) is the seventh most prevalent malignancy and the sixth leading cause of cancer-related death globally, accounting for 509,000 deaths yearly (*Bray et al., 2018*). Despite fast advances in medical technology, EC incidence is expected to rise (*Napier et al., 2014*). The 5-year survival rate for EC patients has seen very modest improvement over the last five decades. Specifically, it has increased from a mere 4% in the 1950s to 17% in 2010 (*Baiu and Backhus, 2020*). The significance of early detection and immediate symptomatic intervention in enhancing EC survival rates necessitates developing a straightforward, effective, and non-invasive method for diagnosing the disease, consequently facilitating its screening process. Imaging techniques, tissue biopsies, and serum tumor markers, including carcinoembryonic antigen (CEA) and CA-199, are now the primary tools for identifying esophageal malignancies (*Siegel et al., 2017*; *Kosugi et al., 2004*). Despite their limited sensitivity in detecting small lesions, imaging techniques remain the prevailing method for cancer diagnosis and surveillance, posing challenges in the early detection of recurrent locations (*Becker et al., 2013*). Similarly, serum biomarkers, including CEA and squamous cell carcinoma antigen, have low sensitivity and specificity for early diagnosis or recurrence detection (*Xu et al., 2014*). Therefore, more potent diagnostic markers for EC are urgently required to enhance diagnosis and non-invasive approaches.

Exosomes, which are small vesicles enclosed by a double membrane and measuring between 40 and 200 nm in diameter, were initially discovered in sheep reticulocytes by *Pan and Johnstone, 1983*; *Kalluri and LeBleu, 2020*. These vesicles are produced through the endosomal pathway and are expelled into the extracellular space (*Latifkar et al., 2019*; *Harding et al., 2013*). Exosomes, extracellular vesicles, are produced by many cells, such as blood, immune, cancer, and stem cells (*Kimiz-Gebologlu and Oncel, 2022*; *Wei et al., 2022*; *Xu et al., 2022*). These exosomes can be secreted into different physiological fluids, including blood, urine, breast milk, ascites, amniotic fluid, saliva, and cerebrospinal fluid (*Ha et al., 2016*; *Li et al., 2014*; *Lässer et al., 2011*). The tumor cell molecular markers are enriched in exosomes transferred between the tumor and normal cells (*Attaran and Bissell, 2022*). Protein expression in tumor patient-derived small extracellular vesicles (sEVs) differs significantly from healthy donors (HD). Accordingly, these differentially expressed sEVs proteins emerge as potential tumor diagnosis biomarkers (*Huang and Deng, 2019*).

Intercellular adhesion molecule-1 (ICAM-1), also known as CD54, is a transmembrane glycoprotein in the immunoglobulin superfamily that is present in a wide range of cell types and is up-regulated in response to a variety of inflammatory mediators. Recent research on acute myeloid leukemia discovered a correlation between CD54 expression and other differentiation-related molecules (e.g., CD34 and HLA-DR) and a significant correlation to French-American-British Cooperative Group (FAB) morphological classification. Moreover, high CD54 expression was associated with germinal center lymphoma in lymphoproliferative disorders (*Maio and Del Vecchio, 1992*; *Wang et al., 2021*). CD54 is also believed to play an important role in several malignancies. In breast, gastric, and colorectal cancers, increased CD54 expression in the cancer cells has been correlated with more favorable prognosis, suggesting a role of CD54 in enhancement of immune surveillance (*Ogawa et al., 1998*; *Fujihara et al., 1999*; *Tachimori et al., 2005*). Previous studies have shown that CD54 can predict the progression and metastasis of gastric cancer (*Gómez-Gallegos et al., 2023*). *Fang et al., 2016* identified circulating tumor cells as a biomarker of treatment selection and liver metastasis in patients with colorectal cancer. However, exosomal CD54 has not been reported in EC.

Herein, the candidate molecule CD54 in EC tissue-derived exosomes was identified using proteomics and validated in plasma exosomes from 122 EC patients. A series of assays, including cell counting kit-8, adhesion, wound healing, and Matrigel invasion, were performed to investigate EC viability, adhesive, migratory, and invasive capabilities. The results showed that CD54 promoted EC proliferation, migration, and invasion. The findings strongly demonstrate that CD54 is a promising biomarker for EC diagnosis and a key molecule for its development.

## Results

### Exosome identification

The aim of this project is to find effective diagnostic indicators for EC. The overall workflow is presented in *Figure 1A*. The nano-flow cytometry (NanoFCM) results revealed that the isolated exosome diameters ranged from 50 to 200 nm (*Figure 1B*). Exosomes appeared as concave hemispherical bilayer

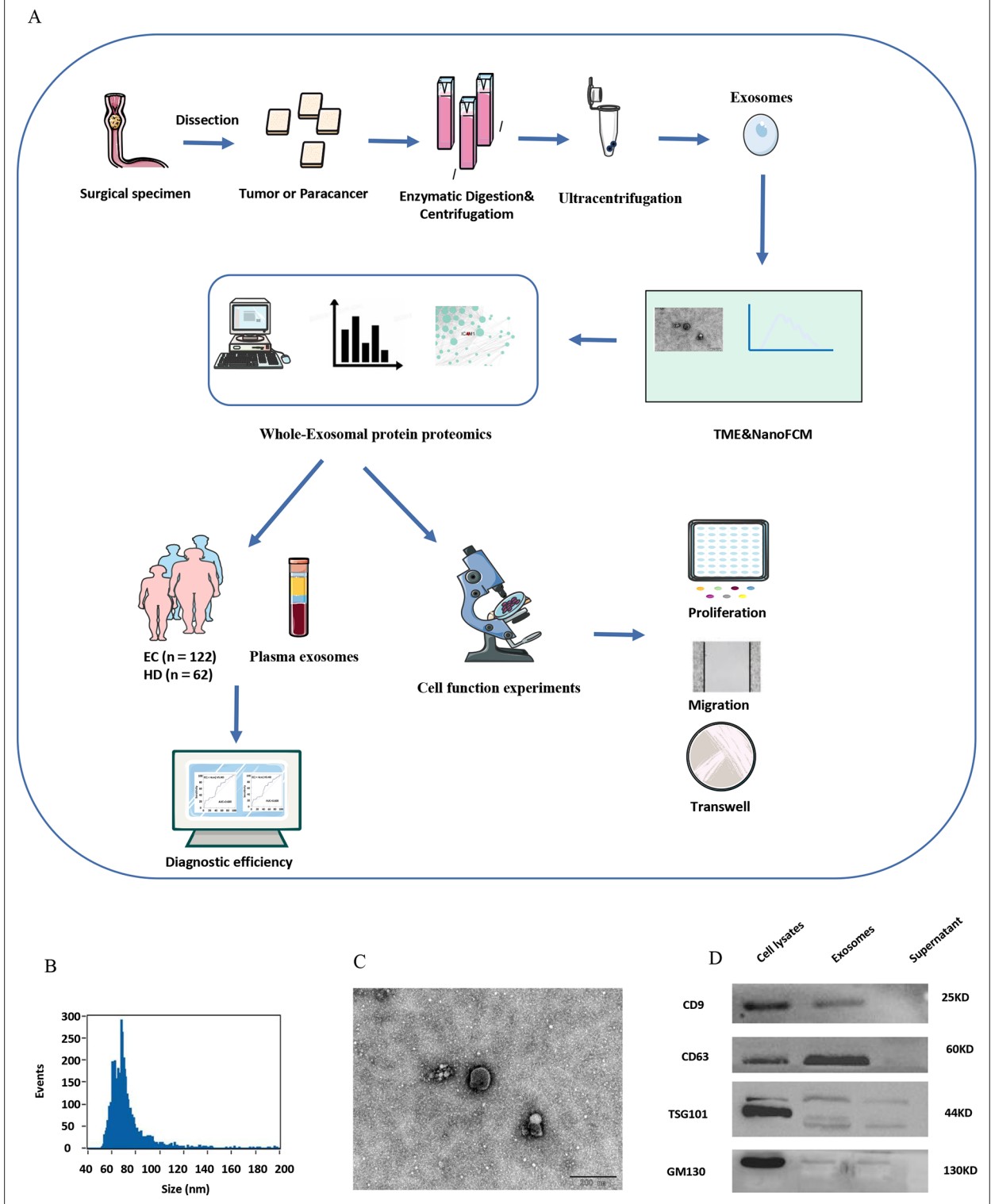

**Figure 1.** Exosome identification. (**A**) Overview of the entire experimental design of this study. (**B**) Nano-flow cytometry (NanoFCM) results show isolated exosome diameters ranging from 50 to 200 nm. (**C**) Transmission electron microscope of exosomes showing round-shaped structures with diameters of approximately 120 nm; scale bar: 100 µm. (**D**) Western blot analysis of CD9, CD63, TSG101, and GM130 exosomal biomarkers.

The online version of this article includes the following source data for figure 1:

**Source data 1.** Original files for the gels in *Figure 1D*.

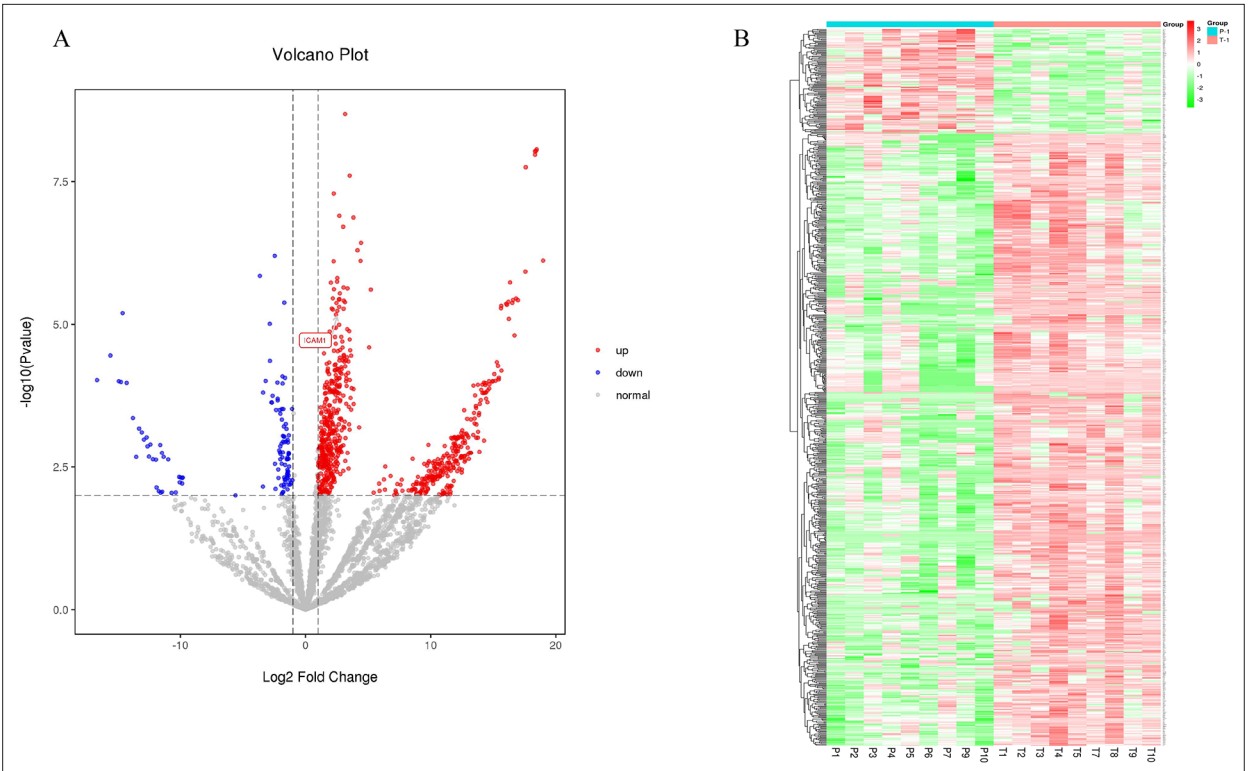

**Figure 2.** Label-free quantitative proteomics profiling of discovery of differentially expressed proteins. (**A**) The volcano plot was drawn using two factors, the fold change (log$_2$) between the two groups of samples and the p-value ($-$log$_{10}$) obtained from the t-test, to show the significance of differences in the data between the two groups of samples. The red and blue dots in the figure denote significantly up-regulated and down-regulated proteins, respectively; the gray dots denote proteins with insignificant differences. (**B**) Differentially expressed protein clustering diagram; columns represent different samples, rows represent different proteins, clustered by log$_{10}$ (protein expression value+1) value, data in each row is normalized, red indicates high expression protein, green indicates low expression protein.

structures with a diameter of approximately 120 nm under a transmission electron microscope (*Figure 1C*). Furthermore, western blot analysis revealed the presence of exosomal protein markers, CD9, CD63, and TSG101, but not a control protein, GM130, in isolated exosomes (*Figure 1D*). These findings are consistent with the definition of exosomes by the International Extracellular Vesicle Society Extracellular Vesicle Society.

## Identification of differentially expressed proteins

Label-free quantitative proteomics profiling was performed on eight cancer and paracancer tissue-derived exosome pairs. The screening criterion to analyze the differentially expressed proteins was as follows: |log2(FC)|≥1.0000000; p≤0.01; fold change (FC) denoted the ratio of expression between two samples (groups). Of the 803 identified proteins, 686 were up-regulated and 117 were down-regulated in EC (*Figure 2A*). Correlation heatmap clustering separated samples from the two groups, indicating a significant difference between cancer proteomes and paracancer tissue-derived exosomes (*Figure 2B*).

## Functional enrichment analysis of differentially expressed proteins

A GO enrichment analysis was performed to reveal the significantly enriched biological functions; p<0.05 denoted statistically significant enrichment results. *Figure 3A* sequentially displays the statistically significant items in biological process (BP), cellular component (CC), and molecular function (MF). These differentially expressed proteins in BPs were mainly related to cellular, single-organism, and metabolic processes. EC-related proteins in MFs were mainly associated with binding, catalytic activity, and structural molecule activity. The differentially expressed proteins in CCs were mainly enriched in cells, cell parts, and organelles, among other components. The KEGG analysis revealed

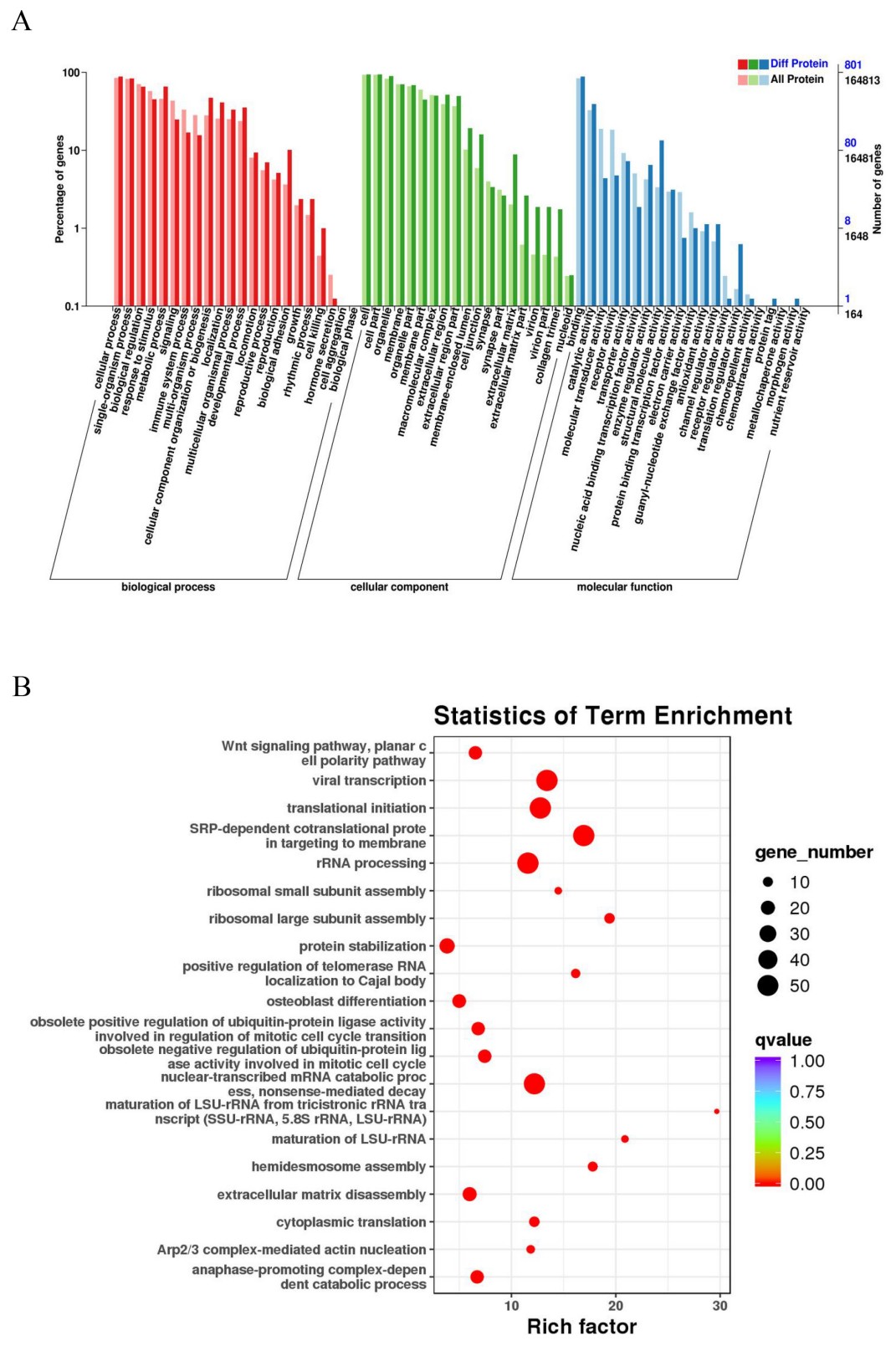

**Figure 3.** GO and KEGG pathway analyses of esophageal cancer (EC)-related proteins. (**A**) Classification of 801 differentially expressed proteins based on biological processes, cellular components, and molecular functions. The horizontal axis is the GO classification, the left side of the vertical axis is the percentage of the number of proteins, and the right side is the number of proteins. (**B**) The abscissa is the enrichment factor; the ordinate is the name of

*Figure 3 continued on next page*

*Figure 3 continued*

the GO term; the size of the dot indicates the number of proteins annotated to this term; and the color indicates the Q value of the significant p-value corrected by multiple hypothesis testing.

that these proteins play a role in the catabolic process of nuclear-transcribed mRNA, viral transcription, translational initiation, and signal recognition particle-dependent cotranslational modification (*Figure 3B*).

## CD54 expression is up-regulated in EC tissues

CD54 expression was assessed by immunohistochemistry in the eight sequenced tissue sample pairs. The results revealed that CD54 was highly expressed in cancer tissues (*Figure 4A*) and at the gene level (*Figure 4B*). Similarly, western blot analysis of the eight tissue sample pairs demonstrated a higher CD54 expression in cancer tissues than adjacent tissues (*Figure 4C*).

## CD54 expression in plasma-derived exosomes

Exosomes were recovered from the plasma of 122 EC patients and 62 HDs using ultracentrifugation. NanoFCM analysis of CD54 expression in exosomes showed that CD54 levels were significantly higher in plasma-derived exosomes in EC patients than in HDs (*Figure 5A–C*). CD54 levels increased as the disease progressed (*Figure 5D*); CD54 expression decreased following resectioning of the patient's mass (7–10 days) (*Figure 5E*).

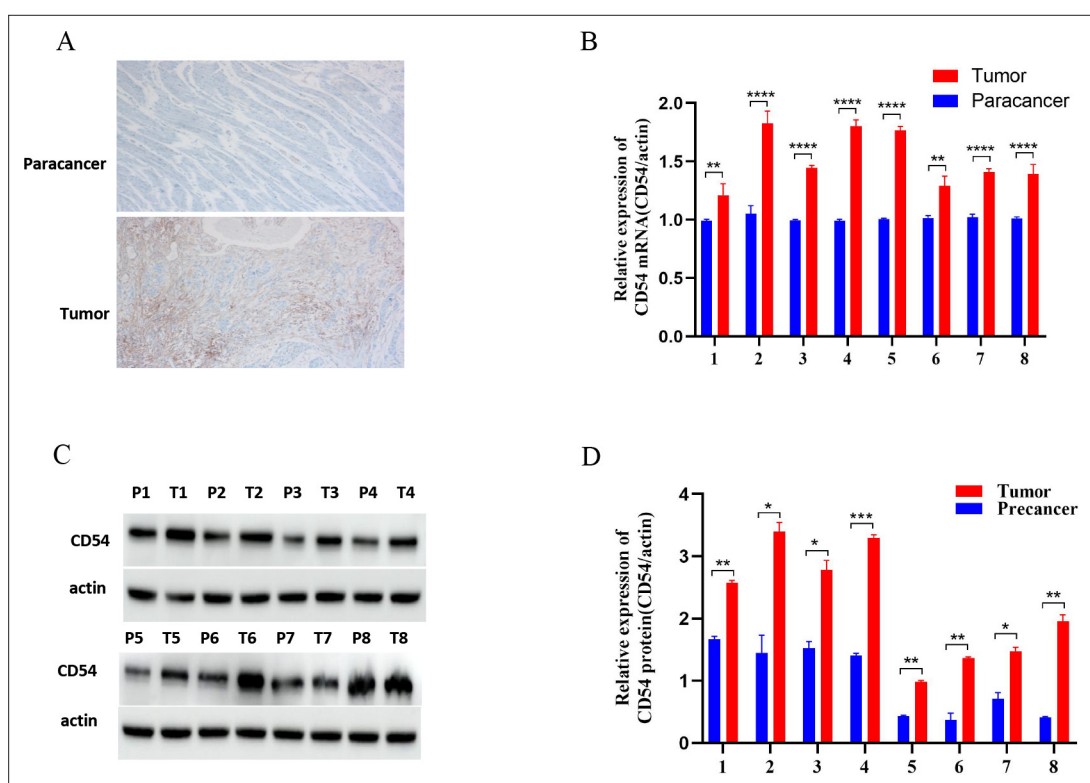

**Figure 4.** Up-regulated CD54 expression in esophageal cancer (EC) tissues. (**A**) Immunohistochemical detection of CD54 expression in EC and paracancer tissues; magnification ×100. (**B**) Real-time quantitative PCR detection of CD54 mRNA expression in eight EC and adjacent tissue pairs. (**C**) Western blot detection of CD54 expression in eight EC and adjacent tissue pairs. (**D**) Quantification analysis of representative western blot images using ImageJ software. *p<0.05; **p<0.01; ***p<0.001; ****p<0.0001.

The online version of this article includes the following source data for figure 4:

**Source data 1.** Original files for the gels in *Figure 4C*.

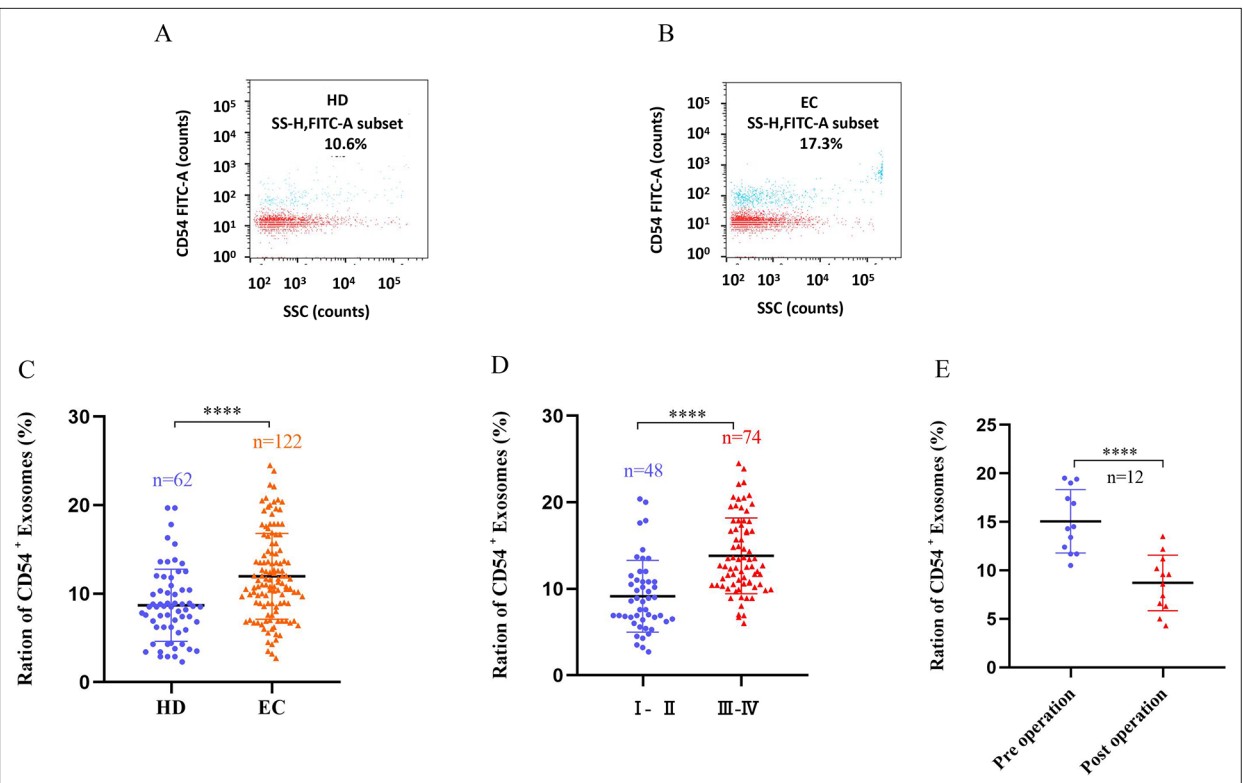

**Figure 5.** CD54 expression in plasma-derived exosomes. (**A–B**) The bivariate dot plots of CD54 FITC fluorescence (y-axis) vs. SSC (x-axis) for exosomes isolated from plasmas of healthy donors (HDs) (**A**) and esophageal cancer (EC) (HCC) (**B**) patients. (**C**) The ratios of CD54+ exosomes vs. total exosomes (CD54+/total) significantly differed between HD and EC groups. (**D**) Exosomal CD54 levels were significantly higher in stage III–IV patients than in stages I–II. (**E**) Changes in plasma exosomal CD54 levels in EC patients (n=19) before (preoperation) and after (7–10 days postoperation) surgical removal of the tumor. ****p<0.0001.

## Correlation analysis between plasma exosomal CD54 levels and clinical characteristics of EC patients

*Table 1* depicts the connection between plasma exosomal CD54 levels and clinical characteristics in EC patients. Plasma exosomal CD54 expression levels were closely related to tumor size (p=0.0308),

**Table 1.** The relationship between plasma exosomes CD54 levels and clinical characteristics of esophageal cancer (EC) patients.

| Clinicopathological characteristics | | N (122) | CD54+ exosomes (%) | p |
|---|---|---|---|---|
| Age (years) | <50 | 49 | 11.56±4.99 | = 0.4490 |
| | ≥50 | 73 | 12.24±4.77 | |
| Gender | Male | 82 | 12.59±4.86 | = 0.3490 |
| | Female | 37 | 11.69±4.83 | |
| Tumor size | <4 cm | 48 | 10.33±4.54 | = 0.0307 |
| | ≥4 cm | 74 | 12.93±4.81 | |
| Distant metastasis | Absent | 50 | 9.86±4.57 | <0.0001 |
| | Present | 72 | 13.38±4.54 | |
| Smoking history | Absent | 49 | 10.11±4.58 | = 0.0004 |
| | Present | 73 | 13.21±4.68 | |
| Drinking history | Absent | 46 | 9.91±4.34 | = 0.0002 |
| | Present | 76 | 13.21±4.71 | |
| Clinical stage | I–II | 48 | 9.13±4.15 | <0.0001 |
| | III–IV | 74 | 13.80±4.38 | |

**Table 2.** Main parameters of receiver operating characteristic (ROC) curve analysis results.

| Variable | AUC | 95% CI | Sensitivity (%) | Specificity (%) | Youden index | p |
|---|---|---|---|---|---|---|
| HD-EC | 0.702 | 0.630–0.767 | 66.13 | 71.31 | 0.3744 | <0.0001 |
| HD-EC (<4 cm) | 0.600 | 0.501–0.694 | 75.81 | 44.44 | 0.2025 | = 0.0733 |
| HD-EC (≥4 cm) | 0.761 | 0.682–0.830 | 66.13 | 81.82 | 0.4795 | <0.0001 |
| HD-EC (stage I) | 0.513 | 0.401–0.625 | 32.26 | 52.38 | 0.1536 | = 0.8631 |
| HD-EC (stage II) | 0.534 | 0.425–0.640 | 20.97 | 92.59 | 0.1356 | = 0.6080 |
| HD-EC (stage III) | 0.830 | 0.74–0.898 | 61.29 | 97.37 | 0.5866 | <0.0001 |
| HD-EC (stage IV) | 0.803 | 0.711–0.877 | 66.13 | 86.11 | 0.5224 | <0.0001 |

distant metastasis (p<0.0001), smoking history (p<0.0004), drinking history (p<0.0002), and clinical stage (p<0.0001), regardless of age (p=0.449) and gender (p=0.349).

## The diagnostic efficiency of plasma exosomal CD54 for determination of EC

The diagnostic efficiency of CD54[+] exosomes in identifying EC was evaluated using receiver operating characteristic (ROC) curve analysis (*Table 2*). The results revealed that diagnostic sensitivity, specificity, and AUC were 66.13%, 71.31%, and 0.702, respectively (*Figure 6A*). Patients with tumor size ≥4 cm performed better in diagnostics than patients with <4 cm (*Figure 6B–C*). These data strongly suggest that early diagnosis of patients is ineffective. Similarly, the assessment of the diagnostic performance of different stages showed 32.26% sensitivity, 52.38% specificity, and an AUC value of 0.513 in stage I (*Figure 7A–D*).

## Reduced CD54 reduces EC proliferation and migration

We used real-time quantitative PCR (RT-qPCR) (*Figure 8A*) and western blot (*Figure 8B–C*) to detect CD54 expression in normal human normal esophageal epithelial cell (HEEC) and EC cells TE-1, OE19, and KYSE-510. The results showed that CD54 was highly expressed in EC cells compared with normal esophageal epithelial cells, especially TE-1 and KYSE-510 (*Figure 8A–C*). Similarly, immunofluorescence results showed that CD54 was highly expressed in TE-1 and KYSE-510 than in HEEC (*Figure 8D*). Next, we used siRNA specific for CD54 (si-CD54) sequences transfected into EC cells, revealing that CD54 expression was reduced (*Figure 8E–F*). Cell viability (*Figure 8G–H*), wound healing (*Figure 8I–J and M–N*), and cell migration (*Figure 8K–L and O–P*) results show that the knocked-down CD54

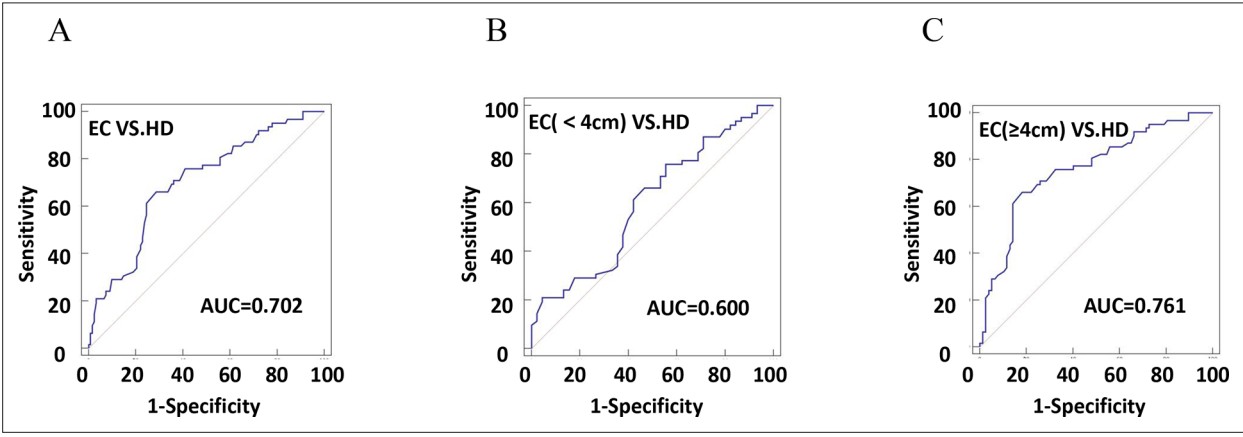

**Figure 6.** The efficiency of plasma exosomal CD54 in esophageal cancer (EC) diagnosis. (**A**) The diagnostic efficiency to distinguish EC patients from healthy donors (HDs). (**B**) The diagnostic efficiency to distinguish EC patients with tumor size <4 cm from HD. (**C**) The diagnostic efficiency to distinguish EC patients with tumor size ≥4 cm from HD.

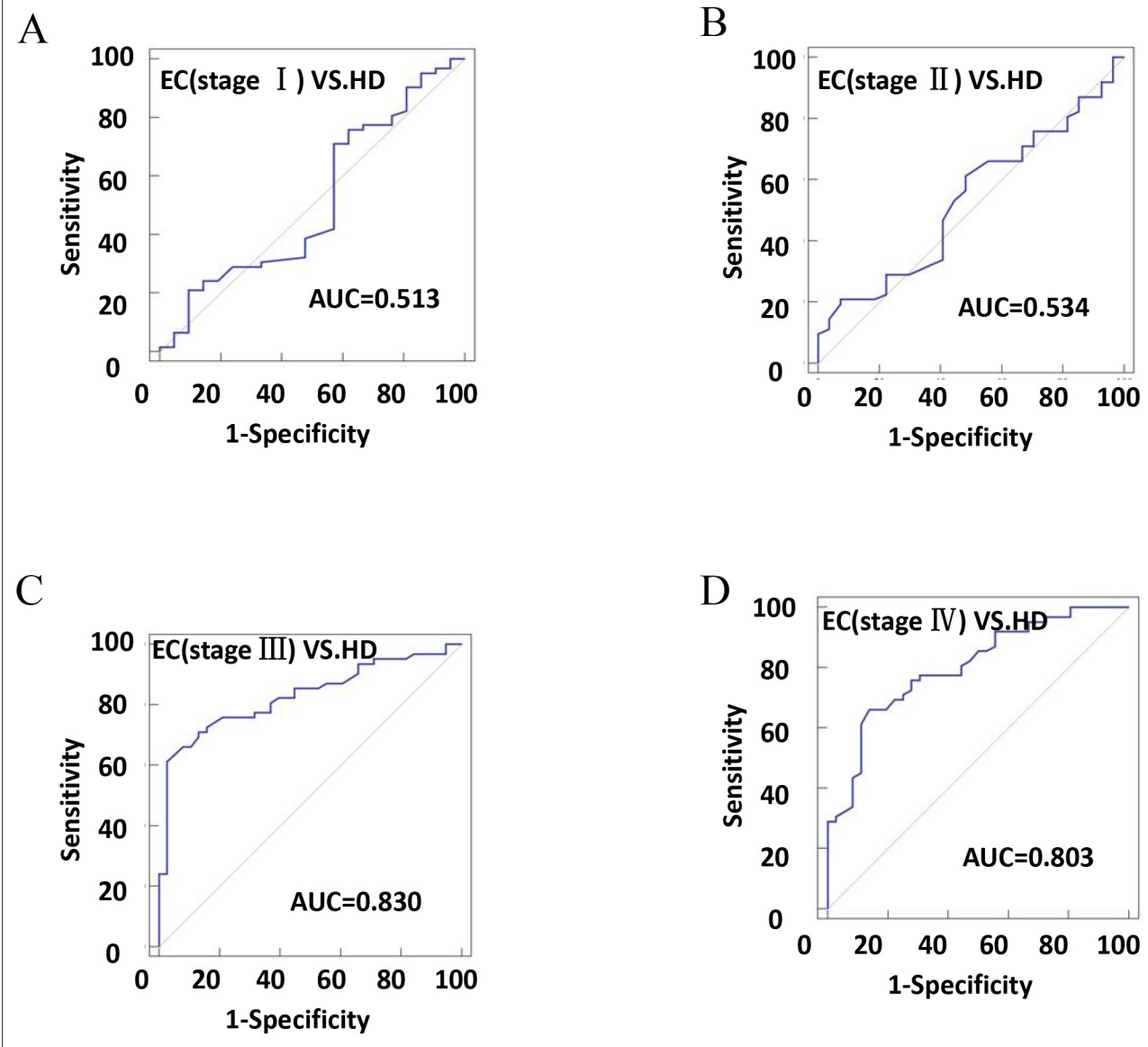

**Figure 7.** The efficiency of plasma exosomal CD54 in esophageal cancer (EC) diagnosis. (**A**) The diagnostic efficiency to distinguish EC patients with stage I from healthy donors (HDs). (**B**) The diagnostic efficiency to distinguish EC patients with stage II from HDs. (**C**) The diagnostic efficiency to distinguish EC patients with stage III from HDs. (**D**) The diagnostic efficiency to distinguish EC patients with stage IV from HDs.

significantly reduced the proliferative and migratory capacities of EC cells TE-1 and KYSE-510. These results strongly suggest that CD54 is important in EC development.

## Discussion

Early cancer detection continues to be of utmost importance in cancer management, given that prompt initiation of clinical interventions is associated with a more favorable outcome (*Whiteside, 2016*). EC is a malignant tumor with a high propensity for metastasis and recurrence. It is commonly diagnosed at later stages, resulting in a poor prognosis. This is mostly due to the delayed onset of clinical symptoms and the absence of early biomarkers for the disease. When distant metastases are detected at the time of diagnosis, the overall 5-year survival rate of EC decreases to less than 5%, as opposed to the 18% survival rate observed in cases where distant metastases are absent (*Jamel et al., 2019*). Endoscopy is widely regarded as the preferred method for detecting and diagnosing BE and EC. However, it is important to note that this approach is not considered cost-effective, practical, or

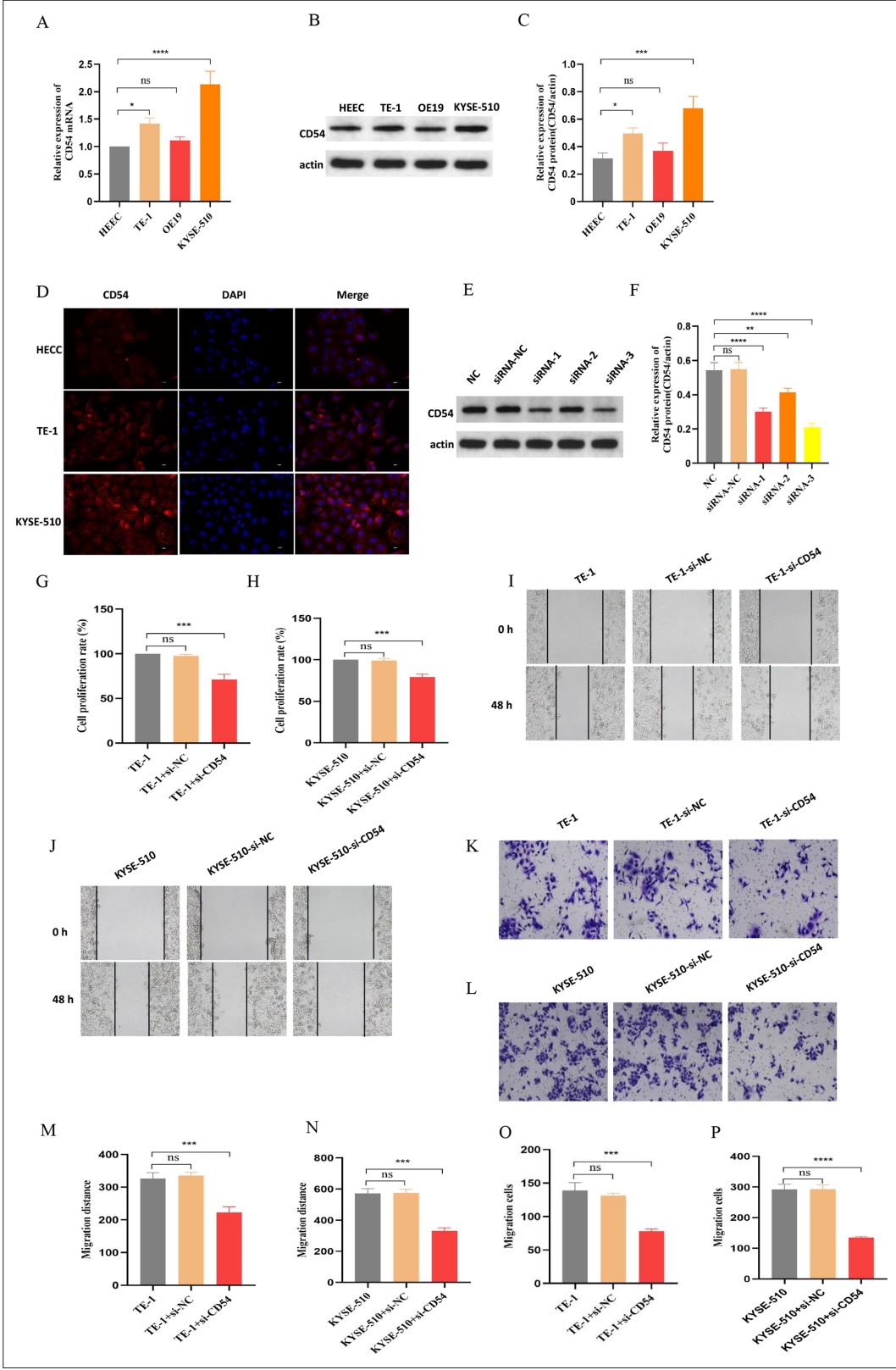

**Figure 8.** Reduced CD54 reduces esophageal cancer (EC) proliferation and migration. (**A**) The relative mRNA expression of CD54 was significantly increased in TE-1 and KYSE-510 cells. (**B**) The protein expression of CD54 was significantly increased in TE-1 and KYSE-510 cells. (**C**) Quantification analysis of representative western blot images using ImageJ software. (**D**) Representative images of CD54 immunofluorescence staining in human normal

*Figure 8 continued on next page*

*Figure 8 continued*

esophageal epithelial cell (HEEC), TE-1, and KYSE-510. Nuclei were stained with DAPI; scale bar: 100 μm. (**E**) The relative protein expression of CD54 was significantly decreased in TE-1 cells using specific siRNA against CD54. (**F**) Quantification analysis of representative western blot images using ImageJ software. (**G–H**) The cell viability was significantly decreased in (**G**) TE-1 and (**H**) KYSE-510 cells using specific siRNA against CD54. (**I–L**) Cell migratory abilities were significantly decreased in (**I, K**) TE-1 and (**J, L**) KYSE-510 cells using specific siRNA against CD54; magnification, ×100. (**M–P**) Histograms showing that CD54 siRNA significantly decreased the migration abilities of (**M–N, K**) TE-1 and (**O–P**) KYSE-510 cells, as demonstrated by the wound healing and cell migration assays, respectively. ns: no significant; **p<0.01; ***p<0.001; ****p<0.0001.

The online version of this article includes the following source data for figure 8:

**Source data 1.** Original files for the gels in *Figure 8B*.

**Source data 2.** Original files for the gels in *Figure 8E*.

non-invasive for screening purposes. Therefore, it is imperative to implement screening measures that facilitate early detection and diagnosis to mitigate the death rate among EC patients.

Herein, of the eight pairs of EC and paracancer tissues exosomes collected for proteomic analysis, we discovered 803 differentially expressed proteins; 686 were up-regulated, and 117 were down-regulated. CD54, intracellular adhesion molecule-1, is an Ig superfamily surface receptor found in leukocytes, endothelial cells, and smooth muscle cells (*van de Stolpe, 1996*); it is an adhesive and co-stimulatory molecule with five extracellular Ig-like domains. Proteomics indicated a substantial increase in CD54 expression in EC-derived exosomes. CD54 was identified as an up-regulated candidate for subsequent investigation. Immunohistochemistry, RT-qPCR, and western blot assessment of C54 expression in EC-derived exosomes yielded comparable results to proteomics. Moreover, NanoFCM evaluation of plasma-derived exosomes demonstrated higher CD54 expression in 122 EC patients than in HDs and increased with disease progression. Notably, low CD54 expression was detected in 12 tumor-resected patients. Further analysis of the relationship between plasma exosome CD54 levels and clinical characteristics of EC patients revealed that CD54 expression was independent of patient gender and age but was associated with tumor size, distant metastasis, smoking history, drinking history, and clinical stage. Evaluation of the diagnostic performance of exosomal CD54 in EC yielded the following results: sensitivity, 66.13%; specificity, 71.31; AUC value, 0.702. Nevertheless, the efficacy of CD54 in the early diagnosis of EC was insignificant. These specific characteristics of patient cohorts in different regions, sample sizes, and potential confounders affect diagnostic accuracy. Therefore, it is necessary to expand the sample size for further study. Additionally, we conducted a functional study of CD54 on EC cells, revealing that CD54 expression of TE-1 and KYSE-510 in EC cells was higher than that in normal HEEC. CD54 knocked down significantly reduced EC proliferation and migration. These results indicate that CD54 plays a key role in EC development.

Through proteomics, we discovered high CD54 expression in EC tissue-derived exosomes; this finding was subsequently validated using patient tissue samples and multiple plasma samples. The presented data indicate that exosomal CD54 holds promise as a potential diagnostic marker for EC. However, the sample size in this study was limited, and further investigation with a larger number of samples is necessary to establish the effectiveness of exosomal CD54 in EC diagnosing. Furthermore, experimental investigations on cellular function have demonstrated that CD54 plays a crucial part in EC development. This study offers a foundation for identifying novel targets for EC diagnosis and therapy.

## Materials and methods

### Sample collection

This study investigated EC patients in the First Affiliated Hospital of Gannan Medical College between February 2021 and July 2021. Eight pairs of surgically resected cancer and paracancer tissues identified as having EC besides plasma from 122 patients and 62 HDs were collected. CT examinations were performed on all patients, and clinical signs, symptoms, and features were obtained from electronic medical records. Patients and HDs completed informed consent forms. This study followed the Declaration of Helsinki and was approved by the First Affiliated Hospital of Gannan Medical College local ethics committee.

## Cell lines and culture

HEEC was purchased from the BeNa Culture collection Co., Ltd (Suzhou, China). TE-1 was purchased from the Fenghui Biotechnology Co., Ltd (Changsha, China), OE19 and KYSE-510 were purchased from the Procell Life Science & Technology Co., Ltd (Wuhan, China). All cells were identified by short tandem repeat (STR) and tested negative for mycoplasma. STR appraisal certificates are uploaded in the supplementary materials. Cells were cultured in Dulbecco's modified eagle medium (DMEM) (Biological Industries) containing 10% fetal bovine serum (FBS; Biochrom, Co., Ltd.) and 1% penicillin/ streptomycin (Invitrogen; Thermo Fisher Scientific, Inc) in a humidified 5% $CO_2$ at 37°C.

## RNA interference

Small interfering (si)RNA specific for CD54 sequences and scrambled negative control (NC) siRNA with CD54 analog nonsense sequences (Sangon Biotech Co., Ltd.) were as follows: CD54 (sense: 5'-GCCAGCUUAUACACAAGAATT'; antisense: 5'- UUCUUGUGUAUAAGCUGGCTT-3') and NC (sense: 5'-UUCUCCGAACGUGUCACGUTT-3'; antisense: 5'- ACGUGACACGUUCGGAGAATT –3'). Cells were seeded into six-well plates and cultured to 80% confluence. The cells were then transfected with 50 nmol/L of NC siRNA and 100 nmol/L of CD54 for 24 hr at 37°C using Lipofectamine 2000 (Invitrogen; Thermo Fisher Scientific, Inc) following the protocols. Briefly, cells were divided into two groups: A blank control (CTRL) group (only treated with serum-free medium), an NC group (transfected with NC siRNA), and a 100 nmol/L CD54 group. All subsequent experiments were performed 24 hr post-transfection.

## Exosome isolation

Tissue exosome isolation: The tissue was cut into 1–2 mm pieces, placed in 2 mL of Roswell Park Memorial Institute 1640 (RIPA-1640) (2 mL of collagenase, 40 U/mL of deoxyribonuclease I), and incubated in a shaker at 37°C for 30 min. The culture supernatant was filtered through 70 µm and centrifuged at 300 × $g$ for 10 min. The supernatant was re-centrifuged at 2000 × $g$ for 20 min. The resultant supernatant was collected, re-centrifuged at 16500 × $g$ for 20 min, and filtered through a 0.22 µm filter membrane. The filtrate was centrifuged at 110,000 × $g$ for 42 min, and the supernatant was discarded. The precipitate was suspended in 1 mL phosphate buffer saline (PBS). The exosome fraction was collected using a size exclusion column and condensed to 100 µL using a 100 kD ultrafiltration tube. All centrifugations were performed at 4°C.

Plasma exosome isolation: Blood was collected in an EDTA anticoagulant tube and centrifuged at 1500 × $g$ for 20 min to remove cells and debris. The supernatant was collected and centrifuged at 3000 × $g$ for 15 min to collect the plasma. If the exosomes did not separate immediately, the separated plasma was stored at –80°C for subsequent processing. Subsequently, 400 µL of the above plasma was transferred into a 2 mL Eppendorf tube, diluted with 1.6 mL PBS, and centrifuged at 10,000 × $g$ for 30 min. The resultant supernatant was collected and centrifuged at 110,000 × $g$ for 64 min, and the supernatant was then gently aspirated. The pellet was resuspended in PBS and centrifuged at 110,000 × $g$ for 64 min. The supernatant was gently aspirated, and the pellet was resuspended in 100 µL PBS to obtain exosomes. The isolated exosomes were stored at –80°C for later applications. All centrifugations were performed at 4°C.

## Trypsin treatment and mass spectrometry analysis

The exosome protein was quantified with a Pierce BCA protein quantification kit (Thermo Fisher Scientific, Inc). Each protein sample received 3 µL of 1 µg/µL of trypsin and 500 µL of 100 mM triethyl ammonium bicarbonate buffer, followed by overnight digestion at 37°C. The digested sample was combined with an equal volume of 1% formic acid and centrifuged at 12,000 × $g$ for 5 min at room temperature. The supernatant was gradually loaded onto a C18 desalting column, washed three times with 1 mL of washing solution (0.1% formic acid and 4% acetonitrile), and eluted twice with 0.4 mL of elution buffer (0.1% formic acid and 75% acetonitrile). The eluents were combined and lyophilized. The lyophilized protein powder was dissolved in 10 µL of 0.1% formic acid in water (solvent A) and injected into a homemade C18 Nano-Trap column (2 cm×75 µm, 3 µm). Peptides were separated in a homemade analytical column (15 cm×150 µm, 1.9 µm) with a mobile phrase of 0.1% formic in 80% acetonitrile (solvent B). The sample was eluted at 600 nL/min flow rate, with the concentration of solvent B increasing from 6% to 100% in 60 min. The peptides were analyzed using a Q Exactive HF-X

mass spectrometer (Thermo Fisher Scientific, Inc) equipped with a Nanospray Flex (ESI) ion source and a spray voltage of 2.3 kV. Following GO analysis, the Clusters of Orthologous Groups databases and KEGG were used to annotate the protein family and pathway.

## Western blot

Proteins from tissues were extracted using tissue lysates. Proteins were separated using 10% SDS-PAGE and transferred to a polyvinylidene fluoride membrane (Millipore). The membrane was blocked with 5% skim milk and incubated overnight with an anti-ICAM1 antibody (Abcam) at 4°C. The membrane was then incubated with a secondary antibody (1:5000) for 1 hr. The protein band was visualized using a fluorescent kit (P0018S, Beyotime) and a chemiluminescence imaging system (T-4600, Tanon). Protein bands were quantified with ImageJ; samples were normalized to β-actin protein levels.

## Real-time quantitative PCR

Total RNA was extracted from EC and paracancer tissues using TRIzol reagent (Thermo Fisher Scientific, Inc). Then, 5 μg RNA was reverse transcribed to complementary DNA (cDNA) using a RevertAid First Strand cDNA synthesis kit (Thermo Fisher Scientific, Inc). qPCR was performed on an ABI 7500 Real-time PCR system (Thermo Fisher Scientific, Inc) using FastStart Universal SYBR-Green Master (Thermo Fisher Scientific, Inc), following the protocol. The thermocycling conditions were as follows: 95°C denaturation for 15 min, followed by 40 cycles of 95°C for 10 s and 60°C for 30 s. The relative mRNA expression levels of target genes were normalized to β-actin and calculated using the $2^{-\Delta\Delta Ct}$ method. The specific sequences of primers were as follows: CD54 forward: 5'-ATGCCCAGACATCTGT GTCC-3', reverse: 5'-GGGGTCTCTATGCCCAACAA-3'; β-actin forward: 5'-GTGGACATCCGCAAAG AC-3', reverse: 5'-GAAAGGGTGTAACGCAACT-3'. All experiments were performed in triplicate.

## Immunohistochemistry

CD54 tissue expression was evaluated using a Strept Actividin-Biotin Complex of immunohistochemistry techniques (SABC kit, Bostere Biotech Company) and the SABC kit protocol. A rabbit anti-human CD54 monoclonal antibody (1:100) was used as the primary antibody. The positive staining sections provided by the antibody kit served as a positive control, while the first antibody was replaced with the same PBS volume as an NC.

## Nano-flow cytometry

NanoFCM (N30E) was used to examine particle concentration, particle size distribution, and surface proteins of exosome samples. Exosomal CD54 expression was evaluated using NanoFCM. Exosomes (50 μL) were mixed with 5 μL of Alexa Fluor 488 anti-human CD54 antibody (Biolegend) and incubated at 37°C for 30 min. Then, exosomes were washed twice by centrifugation at 4°C for 30 min at 110,000 × *g*. The exosome pellet was resuspended in 50 μL of PBS after the final wash, and the CD54-positive exosome proportion was determined using NanoFCM.

## Immunofluorescence staining

The cells were fixed in 4% paraformaldehyde/PBS for 15 min at −20°C and then washed three times using PBS containing 0.1% Triton X-100 (PBST) for 15 min at room temperature. Following blocking in PBS containing 5% BSA for 30 min at 37°C, cells were incubated with the primary antibodies rabbit anti-CD54 (Proteintech; 1:100) for 1 hr at room temperature. The labeled cells were washed three times with PBST and then incubated with Alexa Fluor 594 (goat anti-rabbit IgG; Abcam; 1:100) for 1 hr at room temperature in darkness. After washing three times with PBST, the cell nuclei were stained with DAPI (1 μg/mL, Abcam) for 10 min at room temperature in darkness and detected with a confocal laser scanning microscope (LSM 880; Zeiss AG) at ×63 magnification.

## Cell viability assay

Cell viability was determined by cell counting kit (CCK)-8 assay (HYCEZMBIO) per the manufacturer's instructions. Briefly, EC cells with or without transfection of si-CD54 were incubated in a 96-plate for 24 hr at 37°C. Then, cells were added to 10 μL CCK-8 staining reagent and incubated for another 4 hr at 37°C. Absorbance at 450 nm was measured using a microplate reader (Thermo Fisher Scientific, Inc).

## Wound healing assay

Cell migratory abilities were detected using a wound healing assay. The prepared cells treated with or without si-CD54 were cultured in a six-well plate to approximately 90% confluence using a serum-free medium. The confluent monolayer cells were scratched gently with a 100 μL pipette tip, and images were captured using an inverted microscope (Nikon; magnification, ×100) at 0 and 24 hr. The wound distance was quantified using ImageJ software.

## Cell migration assay

Cell migration assay was performed using 24-well transwell plates with 8 μm pore filters (FALCON) at 37°C for 24 hr following the manufacturer's protocols. Briefly, $1×10^5$ cells were suspended in 100 μL of serum-free medium and added to the upper chamber, and 600 μL DMEM supplemented with 20% FBS was loaded into the lower chamber. After 24 hr of incubation, the non-invaded cells in the top chamber were removed with a cotton swab, and the cells were fixed with 100% methanol for 20 min at room temperature and stained with 0.1% crystal violet (Solarbio Science & Technology Co., Ltd.) for 30 min at room temperature. The stained cells were imaged under an inverted light microscope (Nikon; magnification, ×100) and analyzed using ImageJ software.

## Statistical analysis

GraphPad Prism 8.0 software was employed for statistical analysis, and all quantitative data were presented as the mean±SE. The t-test was used for statistical analysis between the two groups while using one-way analysis of variance (ANOVA) for multiple groups. Nano-flow scatter plot was processed using Flowjo 10.6. The ROC curve was plotted using MedCalc 18.0. $p < 0.05$ denoted statistical significance.

## Acknowledgements

This work was supported by funds from Science and Technology Research Project of Jiangxi Provincial Education Department, China (No. GJJ2201445). We thank Dr. Huang and Dr. Shen of Laboratory Medicine, the First Affiliated Hospital of Gannan Medical University for providing technical guidance and support.

## Additional information

### Funding

| Funder | Grant reference number | Author |
| --- | --- | --- |
| Jiangxi Provincial Department of Science and Technology | No. GJJ2201445 | Zhonghong Lai |

The funders had no role in study design, data collection and interpretation, or the decision to submit the work for publication.

### Author contributions

Dingyu Rao, Conceptualization, Data curation, Writing – original draft, Writing – review and editing; Hua Lu, Data curation, Formal analysis; Xiongwei Wang, Formal analysis, Investigation; Zhonghong Lai, Formal analysis, Methodology; Jiali Zhang, Resources, Data curation, Investigation; Zhixian Tang, Conceptualization, Data curation, Formal analysis, Investigation, Validation

### Author ORCIDs

Dingyu Rao ⓘD https://orcid.org/0000-0002-2182-2682
Hua Lu ⓘD https://orcid.org/0009-0005-7776-0766

### Decision letter and Author response

Decision letter https://doi.org/10.7554/eLife.86209.sa1
Author response https://doi.org/10.7554/eLife.86209.sa2

## Additional files

### Supplementary files
• MDAR checklist

### Data availability
All data generated or analysed during this study are included in the manuscript and supporting file.

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
