## [Editor Report]

This study advances our understanding of the predictive role of tissue-derived biomarkers for esophageal cancer. The evidence supporting the conclusions is solid. However, there are a few areas in which the article may be improved through further analysis and validation of the clinical usefulness of CD54 as diagnostic biomarkers for esophageal cancer. The work will be of broad interest to clinicians, medical researchers and scientists working in esophageal cancer.

---

## [Decision Letter]

**Decision letter after peer review:**

Thank you for submitting your article "Tissue-derived exosome proteomics identifies promising diagnostic biomarkers for esophageal cancer" for consideration by *eLife*. Your article has been reviewed by 2 peer reviewers, and the evaluation has been overseen by a Reviewing Editor and Caigang Liu as the Senior Editor. The reviewers have opted to remain anonymous.

Essential revisions:

1) The manuscript lacks an in-depth description and discussion of the results. The authors should elaborate in the Discussion section.

2) The authors need to conduct a cellular functional assay to clarify how CD54 can impact the occurrence and development of esophageal cancer.

3) The manuscript needs extensive language editing as well.

*Reviewer #1 (Recommendations for the authors):*

Comparative proteomic analysis of exosomes derived from cancerous and paracancer tissues in eight pairs of patients with esophageal cancer was performed to identify the candidate protein CD54. In addition, the study demonstrated that CD54 is a potential diagnostic marker for esophageal cancer. The study makes sense.

1. The manuscript lacks a visual flow chart of the entire research, which is necessary to make it easier for readers to understand.

2. The discussion part is not deep enough, and more content is needed to explore the significance of this research.

3. The writing of the whole manuscript still needs to be revised, focussing on areas such as grammar, and some other writing mistakes.

*Reviewer #2 (Recommendations for the authors):*

This study provides a new idea for liquid biopsy of esophageal cancer. As the information carrier of cell secretion, exosomes can well reflect the changes in the tumor microenvironment. And compared to tissue biopsy, liquid biopsy has the advantages of repeatability and being non-invasive. However, this study only focuses on diagnostic value, and the mechanism of action of extracellular vesicle CD54 on esophageal cancer cells is still unclear.

1. The inclusion and exclusion criteria for collecting plasma from patients were not reflected in the manuscript.

2. The discussion in the article did not provide a more in-depth description of the experimental results, which needs to be supplemented.

3. If researchers can complement the cell function assays, it will make the experimental results more convincing to further clarify how CD54 influences the occurrence and development of esophageal cancer.

---

## [Author Response]

Essential revisions:1) The manuscript lacks an in-depth description and discussion of the results. The authors should elaborate in the Discussion section.

We thank the Editor for this suggestion. We have added some content to the Discussion section that we believe can make the article more engaging.

2) The authors need to conduct a cellular functional assay to clarify how CD54 can impact the occurrence and development of esophageal cancer.

We truly appreciate that the Editor point out this problem. We spent close to three months adding the specific functions of CD54 in EC, which are described in the "Reduced CD54 reduces EC proliferation and migration" section of the manuscript (Figure 8).

3) The manuscript needs extensive language editing as well.

We thank the Editor for this suggestion. We had a professional make changes to our manuscript voice. We believe the quality of the manuscript has been substantially improved.

Reviewer #1 (Recommendations for the authors):Comparative proteomic analysis of exosomes derived from cancerous and paracancer tissues in eight pairs of patients with esophageal cancer was performed to identify the candidate protein CD54. In addition, the study demonstrated that CD54 is a potential diagnostic marker for esophageal cancer. The study makes sense.1. The manuscript lacks a visual flow chart of the entire research, which is necessary to make it easier for readers to understand.

We thank the Reviewer for this suggestion. We added a visual flowchart of the entire study to the manuscript (figure1 A).

2. The discussion part is not deep enough, and more content is needed to explore the significance of this research.

We thank the Editor for this suggestion. We have added some content to the Discussion section that we believe can make the article more engaging.

3. The writing of the whole manuscript still needs to be revised, focussing on areas such as grammar, and some other writing mistakes.

We thank the Editor for this suggestion. We had a professional make changes to our manuscript voice. We believe the quality of the manuscript has been substantially improved.

Reviewer #2 (Recommendations for the authors):This study provides a new idea for liquid biopsy of esophageal cancer. As the information carrier of cell secretion, exosomes can well reflect the changes in the tumor microenvironment. And compared to tissue biopsy, liquid biopsy has the advantages of repeatability and being non-invasive. However, this study only focuses on diagnostic value, and the mechanism of action of extracellular vesicle CD54 on esophageal cancer cells is still unclear.1. The inclusion and exclusion criteria for collecting plasma from patients were not reflected in the manuscript.

We thank the Reviewer for this suggestion. CT examinations were performed on all patients, and clinical signs, symptoms, and features were obtained from electronic medical records. All candidates were excluded from other diseases.

2. The discussion in the article did not provide a more in-depth description of the experimental results, which needs to be supplemented.

We thank the Editor for this suggestion. We have added some content to the Discussion section that we believe can make the article more engaging.

3. If researchers can complement the cell function assays, it will make the experimental results more convincing to further clarify how CD54 influences the occurrence and development of esophageal cancer.

We truly appreciate that the Editor point out this problem. We spent close to three months adding the specific functions of CD54 in EC, which are described in the "Reduced CD54 reduces EC proliferation and migration" section of the manuscript (Figure 8).